# Cerebrovascular Disease in COVID-19

**DOI:** 10.3390/v15071598

**Published:** 2023-07-21

**Authors:** James E. Siegler, Savanna Dasgupta, Mohamad Abdalkader, Mary Penckofer, Shadi Yaghi, Thanh N. Nguyen

**Affiliations:** 1Cooper Neurological Institute, Cooper University Hospital, Camden, NJ 08103, USA; 2Cooper Medical School of Rowan University, Camden, NJ 08103, USA; 3Department of Neurology, Boston Medical Center, Boston University Chobanian and Avedisian School of Medicine, Boston, MA 02118, USAthanh00@gmail.com (T.N.N.); 4Department of Neurology, Rhode Island Hospital, Brown University, Providence, RI 02912, USA

**Keywords:** coronavirus disease 2019, COVID-19, stroke, cerebral vein thrombosis, intracranial hemorrhage, anticoagulation, mechanical thrombectomy, intravenous thrombolysis

## Abstract

Not in the history of transmissible illnesses has there been an infection as strongly associated with acute cerebrovascular disease as the novel human coronavirus SARS-CoV-2. While the risk of stroke has known associations with other viral infections, such as influenza and human immunodeficiency virus, the risk of ischemic and hemorrhagic stroke related to SARS-CoV-2 is unprecedented. Furthermore, the coronavirus disease 2019 (COVID-19) pandemic has so profoundly impacted psychosocial behaviors and modern medical care that we have witnessed shifts in epidemiology and have adapted our treatment practices to reduce transmission, address delayed diagnoses, and mitigate gaps in healthcare. In this narrative review, we summarize the history and impact of the COVID-19 pandemic on cerebrovascular disease, and lessons learned regarding the management of patients as we endure this period of human history.

## 1. History of COVID-19 and Its Variants

Following the emergence of the novel human coronavirus known as severe acute respiratory syndrome coronavirus 2 (SARS-CoV-2), there has been an explosion of research into the global epidemiologic impact and paradigm shifts in care related to the COVID-19 pandemic. As we move into the third year of the pandemic, with rapidly accumulating literature on the topic, we provide an update to prior reviews [1,2] on the association between COVID-19 and cerebrovascular disease.

Currently having infected more than 750 million persons and accounting for more than 6.9 million fatalities worldwide [3,4], the novel human coronavirus disease 2019 (COVID-19) outbreak was first reported in December 2019, with its first cases in Wuhan, China. Initial manifestations of SARS-CoV-2 infection include respiratory failure and multiorgan dysfunction, with a wide variety of neurological presentations. Neurological manifestations related to COVID-19 range from common symptoms of gustatory and olfactory dysfunction, headache, and dizziness, to more severe neurological complications of cerebral venous thrombosis, seizures, posterior reversible encephalopathy syndrome, and stroke, with a stroke occurring in 1.1–1.5% of patients admitted with COVID-19 according to global reports [5,6].

Over time, new variants of SARS-CoV-2 were identified. Among them, the delta and omicron variants were sequenced in May and November 2021, respectively, and have been implicated in more severe clinical phenotypes of COVID-19. Following the emergence of the delta SARS-CoV-2 variant, there was an increased risk of neurological complications, including stroke and seizure, as well as an increase in anxiety disorders and a heightened risk of early mortality [7,8]. During times when the omicron variant of SARS-CoV-2 was more prevalent, there were similarly high rates of neurological complications, while the hospitalization rates and duration of symptoms decreased when compared to the delta variant [9]. Compared with patients infected with the delta variant, those affected by omicron had a lower prevalence of the prototypical COVID-19 symptoms of fever, loss of smell, and persistent cough. However, omicron-infected individuals had a higher prevalence of sore throat, hoarse voice, and joint and muscle pain.

In addition to the significant concerns regarding the variable mortality and long-term complications of COVID-19, including the post-acute sequelae syndrome (“Long COVID”) [8], there were considerable concerns regarding higher levels of transmissibility (particularly with delta and omicron variants) and resistance of SARS-CoV-2 variants to vaccines under development [10,11]. In indirect comparisons of clinical trial data, vaccine efficacy for the omicron variant was found to be lower than for the delta variant in ChAdOx1 nCoV-19 (AstraZeneca), BNT162b2 (Pfizer/BioNTech), and mRNA-1273 (Moderna) vaccines. Even with subsequent immunizations/boosters, vaccine effectiveness for the delta variant was in the range of 72–95% in comparison to 46–68% for the omicron variant [11].

Given the high risk of thrombotic complications with acute SARS-CoV-2, including ischemic stroke and cerebral vein thrombosis, one might expect a rise in the worldwide incidence rate of ischemic stroke and cerebrovascular disease during the COVID-19 pandemic. However, for reasons that are summarized in the following sections, there was a global decline in stroke diagnoses during the early months of the pandemic, with other epidemiologic and paradigm changes as a consequence of the biological and psychosocial effects of COVID-19.

## 2. Direct and Indirect Relationships between COVID-19 and Cerebrovascular Disease

Never before has a virus been so strongly linked to a heightened risk of acute cerebrovascular disease. The risk of stroke has a known association with many transmissible infections, including those responsible for bronchitis, influenza, *H. pylori*, cytomegalovirus, and many others [12]. The inflammatory response to these infections is thought to trigger inflammation and endothelial dysfunction, culminating in vascular events such as ischemic stroke and myocardial infarction [13]. Among the more common infections, the ongoing human immunodeficiency virus pandemic has been associated with a 60% relative increase in the risk of stroke and grows over time, although the overall incidence of stroke with this virus is low (1.28% over a 5-year period) [14]. Influenza, by contrast, is associated with a small but significant early risk (maximal within the first 15 days of symptoms), which disappears within 2 months [15]. The temporal association between stroke and SARS-CoV-2 is similar to the relationship between influenza and stroke in that there is a high early risk that likely decreases with time. However, the risk of stroke is several-fold greater with SARS-CoV-2 than with influenza [16].

Multiple mechanisms account for the unique association between SARS-CoV-2 and stroke (Figure 1). Some of these include increased thromboxane synthesis with associated platelet activation, rapid turnover of fibrinogen, endothelial dysfunction, and inflammation, as well as thrombus formation following cardiac dysfunction. Following infection, the SARS-CoV-2 spike protein activates platelets via platelet angiotensin-converting enzyme 2 (ACE2) receptors, resulting in heightened expression of platelet integrin αIIbβ3 and P-selectin, which facilitates degranulation and platelet aggregation [17]. The vascular endothelium is also highly susceptible to viremia given its surface expression of ACE2 receptors, which permits viral entry into endothelial cells, leading to activation/disruption [18]. In parallel with these pathways responsible for platelet activation and endothelial dysfunction, SARS-CoV-2 indirectly activates factor X via inflammatory mediators (e.g., IL-6 and IL-8), which increase tissue factor expression, thereby activating the extrinsic pathway [19]. Furthermore, 30–40% of patients with severe COVID-19 may develop myocardial ischemia, elevated troponins, and new heart failure with resultant ventricular dysfunction [20], potentially contributing to intracardiac thrombus formation and stroke or systemic embolism. There is also a suggestion of elevated anticardiolipin IgA, and beta-2-glycoprotein IgG and IgM levels in patients with COVID-19 in several reports [21,22,23], but these serum findings are also found in patients with other acute infections (unrelated to SARS-CoV-2).

Among the multiple thrombotic complications of COVID-19, ischemic stroke has been reported in approximately 1.0–1.5% of all hospitalized individuals who test positive for SARS-CoV-2 [6], with twice as many patients having no identifiable mechanism of cerebral infarction (>40%) as conventional stroke cohorts [24]. Furthermore, in one early multinational cohort of 156 patients with stroke and COVID-19, nearly half (49.5%) presented with a proximal or medium vessel occlusion on initial neuroimaging, a nearly doubled risk compared with historic stroke cohorts with traditional mechanisms [25]. Even more concerning, when considering the “cryptogenic” mechanism of stroke as being directly related to SARS-CoV-2, the risk of early mortality may be five-fold greater than that of patients with other suspected mechanisms of infarction, according to one case–control study (adjusted OR 5.16, 95% CI 1.41–18.87) [26].

In addition to these cerebrovascular complications of SARS-CoV-2 infection, although it has not been explicitly studied, the psychosocial consequences of COVID-19 may also impact the risk of cerebrovascular disease. The early avoidance of healthcare institutions in the setting of milder cerebrovascular events [27], delays (or cancellations) in primary care appointments [28], and other factors may have inadvertently affected the control of vascular risk factors and heightened long-term stroke risk. Moreover, the long-term consequences of COVID-19 include an increased risk of diabetes, congestive heart failure, coronary disease, and hypertension [29], which can directly increase the risk of ischemic stroke. Other long-term consequences of COVID-19 such as fatigue, brain fog, and depression [30], which can indirectly augment stroke risk by influencing activity, diet, and lifestyle preferences. The direct and indirect factors associated with stroke and disability following COVID-19 are illustrated in Figure 1.

## 3. Cerebral Vein Thrombosis

Beyond its effect on the cardiopulmonary system and the vascular endothelium, SARS-CoV-2 has been associated with a higher risk of venous thromboembolism and cerebral venous thrombosis (CVT) [31,32]. These thromboembolic events are not only disabling in and of themselves [33] but they are also thought of as biomarkers of more severe COVID-19 illness with greater morbidity and mortality. The results from one meta-analysis early in the course of the pandemic, including >8000 patients with COVID-19 (21% of whom developed venous thromboembolism), indicated a 74% higher odds of mortality when venous thromboembolism occurred concomitantly with COVID-19 (odds ratio (OR) 1.74, 95% confidence interval (CI) 1.01–2.98) [34].

For a condition with a low historic incidence rate of 1–2 per 100,000 [35], CVT is over 10 times more common in patients infected with SARS-CoV-2 than in patients free of this infection [36,37]. In contrast to classic CVT, which is more common in younger female patients and is generally associated with a favorable prognosis, CVT in patients with COVID-19 infection has been more common in older males without traditional risk factors and with greater morbidity and mortality than CVT associated with other conditions [38,39,40]. Along with other data regarding the thrombotic risk of SARS-CoV-2, the risk of CVT gave rise to early recommendations for empiric anticoagulation among hospitalized patients with COVID-19, although evidence justifying these early recommendations was limited (see Section 4).

The elevated incidence of CVT in hospitalized COVID-19 patients, coupled with their hypercoagulable state, provides compelling evidence of a causal link between COVID-19 infection and CVT. However, despite the fact that many studies have shown an increased incidence of CVT in patients with COVID-19 infections, the true incidence and prevalence of CVT among patients with COVID-19 remain unknown. Furthermore, these epidemiological and comparative studies before and during the COVID-19 pandemic should be interpreted with caution as these incidences have not been estimated from the same population (e.g., CVT incidence was largely calculated among hospitalized COVID-19 patients and not the total COVID-19-positive population), and CVT risk factors may be confounded by known COVID-19 morbidity risk factors. For instance, one large multinational longitudinal cross-sectional study (n = 217,560 COVID-19 patients, n = 2313 CVT patients) showed no significant differences in CVT volume or CVT in-hospital mortality overall between the first year of the COVID-19 pandemic and the pre-pandemic year [41]. Investigators from this study showed, however, that patients with CVT and COVID-19 had higher in-hospital mortality than COVID-19-negative patients (15.0% vs. 4.5%, *p* < 0.01).

CVT has also been reported following certain SARS-CoV-2 vaccinations, mostly with the adenovirus vector-based vaccines (Ad26.COV2.S (Janssen/Johnson & Johnson, Titusville, FL, USA) and ChAdOx1 (AstraZeneca, Cambridge, UK)) [36,42]. The similarities between the clinical syndrome reported in these patients and spontaneous heparin-induced thrombocytopenia led investigators to identify circulating platelet-activating platelet factor 4 (PF4) antibodies in many of these patients [43]. The condition has been named *vaccine-induced immune thrombotic thrombocytopenia* and has preferentially affected young and middle-aged women without pre-existing conditions [44]. These reports have prompted several countries to restrict the use of these vaccines, especially in younger patients. Although the relative morbidity of VITT is concerning, it remains a rare event, and the incidence of CVT among patients hospitalized with COVID-19 is significantly higher than the VITT-related CVT [38]. Using publicly available data from Our World In Data, we have previously estimated that, across various age groups, ChAdOx1 and Ad.26.COV2.S may be associated with >95% relative risk reduction for COVID-19-associated CVT as compared to the VITT-related COVID-19 [36]. For this reason, and for many other systemic and public health reasons, the advantages of SARS-CoV-2 vaccination far exceed this risk of vaccination. Moreover, VITT and venous thromboembolic events have not been demonstrated in mRNA-based vaccines [45]. That said, the choice of one vaccine over another ought to be made at the patient level and considering the individualized risk of thrombotic events.

## 4. Antithrombotic Strategies

Given the high risk of thrombotic events associated with COVID-19, the safety and efficacy of anticoagulation have been explored in a number of randomized clinical trials [46]. More specifically, it has been established that the prothrombotic state of COVID-19 may be mitigated with heparin, which has anti-factor Xa activity, anti-inflammatory effects, and potential antiviral effects against SARS-CoV-2 [47]. Importantly, the benefit of antithrombotic treatment in COVID-19 has not been explored in randomized clinical trials for secondary prevention of ischemic stroke. Furthermore, trials evaluating antithrombotic therapy have varying inclusion criteria, with some restricting eligibility to those with critical illness (e.g., those requiring invasive ventilation or with multiorgan dysfunction), or those with non-critical illness (see Figure 2 for details).

The RAPID clinical trial investigators evaluated the benefit of therapeutic heparin in critically ill patients with elevated D-dimers for the composite outcome of in-hospital mortality, invasive or non-invasive ventilation, or admission to an intensive care unit. The investigators found no significant advantage of anticoagulation over the standard of care; however, mortality was significantly lower with anticoagulation (OR 0.22, 95% CI 0.07–0.65) [48]. The HEP-COVID investigators also reported a benefit of therapeutic anticoagulation in patients with COVID-19 and elevated D-dimers for the outcome of venous or arterial embolism, and death due to any cause (relative risk 0.68; 95% CI 0.49–0.96). In this trial, the benefit was observed exclusively in non-critically ill patients [49]. Unfortunately, the risk of stroke was not reported in either RAPID or HEP-COVID. In the largest randomized clinical trial of non-critically ill patients with COVID-19, the ATTACC, ACTIV-4a, and REMAP-CAP investigators found that therapeutic anticoagulation with heparin increased the probability of patients surviving hospitalization and being discharged with a reduced duration of need for intensive care (median adjusted OR of organ support-free days 1.27, 95% credible interval 1.03–1.58) [50]. Notably, this trial included all comers with COVID-19, and the event rates for primary intracerebral hemorrhage were zero in both arms, and only three ischemic strokes occurred during hospitalization (two with standard-of-care thromboprophylaxis and one with anticoagulation) [50]. The FREEDOM COVID Trialists also reported a survival advantage with therapeutic anticoagulation, although there was no benefit for the primary trial composite endpoint of all-cause mortality, requirement for an intensive care unit level of care, systemic thromboembolism or ischemic stroke [51].

In a meta-analysis of available trial data, there was a significant benefit of therapeutic anticoagulation for the reduction in major thrombotic events (OR 0.47, 95% CI 0.24–0.90) but only in non-critically ill patients [52]. While there appears to be some benefit of short-term therapeutic anticoagulation with heparin and a potential reduction in the risk of thromboembolism in certain patients with elevated D-dimers in non-critically ill patients, we cannot know whether therapeutic anticoagulation reduces the probability of ischemic stroke or increases the risk of hemorrhagic stroke in these patients. There are currently no ongoing trials investigating the benefit of treating patients with therapeutic anticoagulation post-discharge for patients with an ischemic stroke. In the absence of high-quality data (and with no planned trials in a stroke population with COVID-19), many stroke providers believe it to be reasonable to consider (at least short-term) therapeutic anticoagulation in patients with ischemic stroke and COVID-19, especially if they are found in multiple vascular territories with an elevated D-dimer, suggesting an embolic phenomenon (Figure 2) [53,54]. In these situations, there is no specific D-dimer threshold high enough to warrant anticoagulation (or at which anticoagulation may be of benefit), although studies have identified heightened thrombotic events with D-dimer thresholds ranging from 1 μg/mL [55] to 500 μg/mL [56].

Although there is likely a benefit of anticoagulation in the primary prevention of major thrombotic events in select patients with COVID-19, benefits for the secondary prevention of ischemic stroke in COVID-19 patients remain unclear. Furthermore, guidelines are limited for these scenarios. The 2021 American Heart Association guidelines on secondary stroke prevention (published 15 months after the World Health Organization declared COVID-19 a pandemic) do not mention COVID-19 [57]. Presumably, the risk of subsequent arterial or venous thrombosis following ischemic infarction in a patient with COVID-19 (particularly a non-critically patient with an elevated D-dimer) is high [58,59]. The National Institutes of Health COVID-19 Treatment Guidelines Panel most recently updated their recommendations in December 2022 for the treatment of COVID-19 and thromboembolic disease, but there is no specific mention of secondary stroke prevention in these guidelines [60]. (Many of these recommendations pertain to the treatment of venous thromboembolism.) In patients with arterial ischemic stroke and an indication for therapeutic anticoagulation (e.g., atrial fibrillation), it is reasonable to treat with therapeutic anticoagulation as long as the risk of bleeding is outweighed by the benefit of anticoagulation. For most patients with stroke and COVID-19, but no alternative indication for anticoagulation, single antiplatelet (or short-term dual antiplatelet) therapy is safe and may be effective. For patients with embolic cryptogenic infarcts presumably due to COVID-19 hypercoagulability, particularly those with elevated D-dimers who are non-critically ill, short-term anticoagulation (30–60 days) with heparin and bridge to a direct oral anticoagulant can be considered, followed by antiplatelet monotherapy. For stroke patients with incidentally found SARS-CoV-2 via nasopharyngeal polymerase chain reaction, and no concern for hypercoagulable state, antiplatelet monotherapy (or short-term dual antiplatelet treatment, per guidelines [57]) is reasonable for secondary stroke prevention.

Any potential benefit of antithrombotic therapy must be weighed against the risk of hemorrhage. Among patients with COVID-19, there is a heightened risk of intracranial hemorrhage [61]. An international study showed that patients with COVID-19 and acute ischemic stroke had higher rates of bleeding complications with revascularization treatment (intravenous thrombolysis or mechanical thrombectomy) and worse outcomes than contemporaneous patients who were being treated without COVID-19 [62]. Nonetheless, in the absence of a control group of patients who did not receive revascularization treatment, there are insufficient data regarding the risks of intravenous thrombolysis in acute ischemic stroke [63]. Therefore, COVID-19 (or suspected COVID-19) should not be a contraindication to systemic thrombolysis. Furthermore, endovascular recanalization is not thought to be futile in patients with proximal large vessel occlusion who would otherwise meet the criteria for thrombectomy [2]. While proximal intracranial occlusions are independent risk factors for poor outcomes and early mortality in the setting of COVID-19 [26,64], these patients should not be excluded from endovascular treatment [65]. One multicenter retrospective analysis of 575 patients with proximal intracranial occlusion (n = 194 with COVID-19) reported lower rates of successful recanalization (modified thrombolysis in cerebral infarction grade 3) in patients with COVID-19 versus non-COVID-19 patients (39.2% vs. 67.2%; adjusted OR 0.4, 95% CI 0.2–0.8, *p* < 0.01) with higher rates of discharge with mRS > 2 (79.8% vs. 66.7%; adjusted OR 2.6, 95% CI 1.1–5.8, *p* = 0.03) with propensity score matching [66]. However, these patients still achieved better clinical outcomes than historic cohorts of non-COVID-19 patients with proximal occlusions treated medically.

The duration of antithrombotic treatment in patients with stroke presumably due to COVID-19 is unclear. While the severity of COVID-19 is mediated, in part, by many vascular risk factors (such as diabetes and chronic obstructive pulmonary disease), which predispose to cerebrovascular events, COVID-19 has been implicated as a unique mechanism of cerebral infarction irrespective of this clinical history. For these patients who survive COVID-19 and have no other cause of stroke, it is unclear what—if any—benefit may be gained from lifelong antithrombotic treatment. Such ill-defined benefit from long-term antiplatelet or anticoagulant therapy following COVID-19-associated stroke is much like the unclear advantage of long-term antithrombotic therapy in cervical artery dissection. The 2021 Guidelines from the American Heart Association have recommended anticoagulation for patients in a hypercoagulable state [57]. However, this is dependent on the cause of the patient’s hypercoagulable state, and it is still unclear whether oral anticoagulation with direct oral anticoagulants or vitamin K antagonists offers differential protection from thrombotic events when compared to heparin.

The reader is referred to the National Institutes of Health [60] for the latest updates on antithrombotic recommendations in COVID-19 and to be cognizant of any future societal recommendations for the treatment of COVID-19-associated cerebrovascular disease. As with any off-label treatment, deviations from such recommendations or guidelines ought to be carefully considered and justified given the rapidly changing evidence for treatment.

## 5. COVID-19 Impact on Stroke Systems of Care

For many reasons, the COVID-19 pandemic has led to a decline in new diagnoses of acute cerebrovascular disease [5,6,27,67,68], myocardial infarction [69,70], and other acute medical conditions [71]. Some of this epidemiologic shift was driven by the abrupt change in medical-seeking behaviors of patients (who became avoidant of healthcare institutions due to fear of contracting SARS-CoV-2) [72,73,74]. However, there is also compelling evidence that declines in other transmissible infections, such as influenza [75], mediate declines in vascular events [2,12]. Despite the greater risk of ischemic stroke due to SARS-CoV-2 than infections like influenza [16], the significantly lower incidence rate of other communicable respiratory and gastrointestinal infections during the COVID-19 pandemic [75,76] (which may be more common than SARS-CoV-2) likely displaced any rise in the stroke rate due to SARS-CoV-2.

Early in the course of the COVID-19 pandemic, as healthcare systems adapted to contact and respiratory precautions, several major barriers to acute stroke care emerged [1]. First, there was a dramatic change in available emergency medical services with fewer available first responders in the community. Second, while the overall number of patients treated in emergency departments fell steeply [77], safety precautions implemented in emergency departments led to bottlenecks in acute care for many conditions, including stroke. While data do not indicate consistent delays in neurodiagnostic testing (e.g., due to the need for frequent decontamination of equipment), resource limitations and precautions have been associated with delays in intravenous thrombolysis, according to several large analyses [78,79] and mechanical thrombectomy [80], with expected deleterious effects on clinical outcomes [81].

As stated previously, SARS-CoV-2 has been directly related to large vessel intracranial occlusions [26,82], which are amenable to endovascular treatment. Access to endovascular treatment centers and time to recanalization therapy do not seem to have been adversely affected by the COVID-pandemic [83,84], despite the precautions necessary to maintain the safety of patients and providers. Pandemic preparedness campaigns and early guidance from multiple vascular societies may have influenced the stability of treatment times and specialist availability [63,85,86]. In addition, many non-urgent endovascular cases were rescheduled to reduce unnecessary patient exposure in healthcare settings [2,87], with concomitant increases in endovascular treatment (over open surgical treatment) of other acute conditions such as ruptured intracranial aneurysms [67].

In addition to these collateral effects on acute intervention and stabilization of patients with stroke during the COVID-19 pandemic, we also witnessed changes in patient and family education, disposition planning, and post-discharge follow-up. With the implementation of contact precautions, many family members were prohibited from entering healthcare facilities. Therefore, an extensive amount of stroke education was provided to patients in person, or to caregivers via telephone, when possible. While it has not been well studied, the limited bandwidth of providers during this time and these changes in education and counseling may have negatively impacted patient and caregiver teaching. Further compounding the (potentially suboptimal) education difficulties during the COVID-19 pandemic was the reduction in patient time spent with rehabilitation specialists. A large number of stroke [88,89] and non-stroke [90,91] patients admitted during the pandemic who were potentially eligible for discharge to acute inpatient rehabilitation facilities were discharged more frequently to home (with fewer available therapy services). This disposition derailment may be due to a number of collateral effects of the pandemic. First, discharges to home may be preferred by patients, caregivers, and providers because they can be planned more quickly than arranging a discharge to an inpatient facility. Second, discharges to home (which may be faster) can reduce the exposure of patients to hospital and rehabilitation facility staff, as well as other patients, potentially carrying SARS-CoV-2. And finally, during the pandemic, discharges to post-acute care institutions were delayed due to a decline in the availability of rehabilitation and nursing facilities [92]. At follow-up, in-person contact with healthcare providers and hospital staff was quickly restricted. While telemedicine appointments became a new standard of care and reduced the exposure of patients and caregivers to SARS-CoV-2, these virtual services increased the accessibility of patients to specialty providers across great distances. Furthermore, telemedicine reduced the consumption of personal protective equipment [93]. That said, the (practically obligatory) transition from in-person to telemedical visits was fraught with early difficulties regarding reimbursement for services and limitations of physical assessments, many of which have been assuaged over time.

## 6. Future Directions

To say that the COVID-19 pandemic has altered the epidemiology of stroke and transformed the manner in which we care for our patients would be an understatement. As we endure through this pandemic, we would be remiss in ignoring the lessons learned along the way. Our deeper understanding of the relationship between transmissible infections and ischemic stroke should reinforce the value of primary care, attention to basic hygienic interventions, and vaccinations. The success of telemedicine in providing subspecialty care to rural areas and patients with limited mobility or access to providers has the potential to reduce many disparities. And while there is much we have learned for the better, we have also grown more cautious to accept published information as fact. Early reports seeking to dispel myths of stroke due to COVID-19 [94], or misleading statements pertaining to the risks of COVID-19 vaccination [95], have caused us to question social media, journalism, and even scientific, peer-reviewed publications. Furthermore, such misinformation campaigns have sowed a public sense of mistrust of medical providers so deep that it may take a generation (or more) to rectify. With that in mind, let us accept and embrace this journey to rehabilitate the public perspective of healthcare providers and clinical science. As patient advocates, we have a renewed obligation to leverage all resources available to us—ranging from scientific methods to social media—to promote the dissemination of accurate information and improve the well-being of our patients and communities.

## Figures and Tables

**Figure 1 viruses-15-01598-f001:**
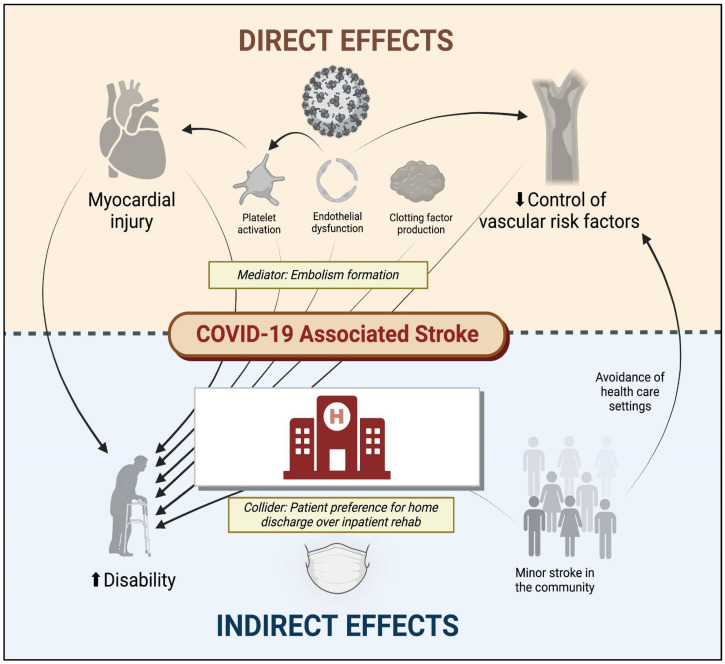
Direct and indirect effects of COVID-19 on cerebrovascular disease. COVID-19 denotes coronavirus disease 2019. Figure generated using biorender.com.

**Figure 2 viruses-15-01598-f002:**
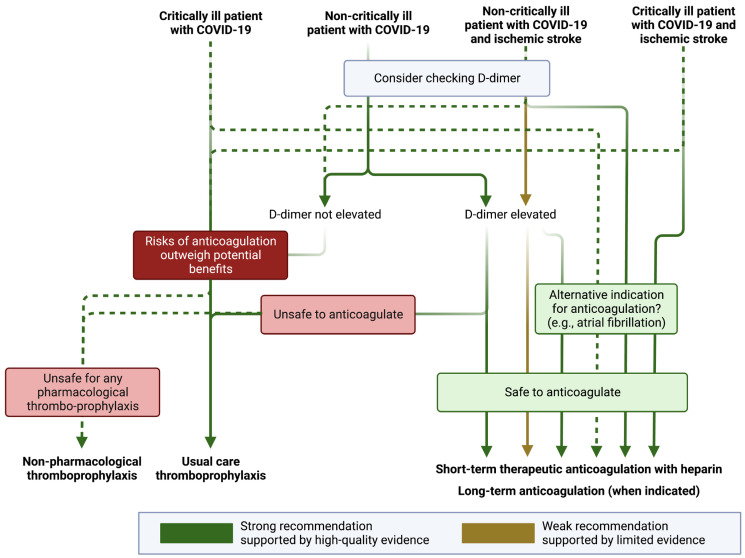
Antithrombotic recommendations in patients with COVID-19. COVID-19 denotes coronavirus disease 2019. Figure generated using biorender.com. Definitions for “critically ill” and “non-critically ill” vary between studies. Generally speaking, “critically ill” patients with COVID-19 are those who require intensive care (e.g., requiring invasive mechanical ventilation, frequent nursing assessment of vital signs, use of vasopressors, or those with multiorgan failure) or are at high risk of imminent deterioration (e.g., those with increasing oxygen requirements or with evidence of new or progressive organ dysfunction). Non-critically ill patients are those who are managed in a non-intensive medical or surgical unit or those well enough to remain outpatient.

## Data Availability

Not applicable.

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
