# Peer review of "Cerebrovascular Disease in COVID-19"

_viruses, 2023, doi:10.3390/v15071598_

Round 1
Reviewer 1 Report
The manuscript a review, Title" Review 1
Cerebrovascular Disease in COVID-19" submitted, by James E. Siegler and coauthor's.
Before moving to review the manuscript, a graphical abstract is highly recommended, to under in one look.
Being a review manuscript, I would recommend the author's to improve the article by including similar articles, from the link (https://pubmed.ncbi.nlm.nih.gov/?term=Cerebrovascular+in+COVID+19&filter=pubt.review&filter=years.2023-2023&sort=pubdate&size=100)
or some articles are selected or search has been refined, please.
The following articles need the attention of author's:
Menezes, R. G., Alabduladhem, T. O., Siddiqi, A. K., Maniya, M. T., Al Dahlawi, A. M., Almulhim, M. W. A., ... & Almas, T. (2023). Cerebrovascular disease in COVID-19: a systematic review and meta-analysis. Le Infezioni in Medicina, 31(2), 140
Siegler, J. E., Dasgupta, S., Abdalkader, M., Penckofer, M., Yaghi, S., & Nguyen, T. N. (2023). Cerebrovascular Disease in COVID-19.
Owens CD, Pinto CB, Detwiler S, Mukli P, Peterfi A, Szarvas Z, Hoffmeister JR, Galindo J, Noori J, Kirkpatrick AC, Dasari TW, James J, Tarantini S, Csiszar A, Ungvari Z, Prodan CI, Yabluchanskiy A. Cerebral small vessel disease pathology in COVID-19 patients: A systematic review. Ageing Res Rev. 2023 Jul;88:101962. doi: 10.1016/j.arr.2023.101962. Epub 2023 May 22. PMID: 37224885; PMCID: PMC10202464.
Menezes RG, Alabduladhem TO, Siddiqi AK, Maniya MT, Al Dahlawi AM, Almulhim MWA, Almulhim HW, Saeed YAA, Alotaibi MS, Alarifi SS, Alkathiry AM, Almas T. Cerebrovascular disease in COVID-19: a systematic review and meta-analysis. Infez Med. 2023 Jun 1;31(2):140-150. doi: 10.53854/liim-3102-2. PMID: 37283635; PMCID: PMC10241400.
Finsterer J. Neurological Adverse Reactions to SARS-CoV-2 Vaccines. Clin Psychopharmacol Neurosci. 2023 May 30;21(2):222-239. doi: 10.9758/cpn.2023.21.2.222. PMID: 37119215; PMCID: PMC10157009.
Solomon IH, Singh A, Folkerth RD, Mukerji SS. What Can We Still Learn from Brain Autopsies in COVID-19? Semin Neurol. 2023 Apr;43(2):195-204. doi: 10.1055/s-0043-1767716. Epub 2023 Apr 6. PMID: 37023787.
Sonneville R, Dangayach NS, Newcombe V. Neurological complications of critically ill COVID-19 patients. Curr Opin Crit Care. 2023 Apr 1;29(2):61-67. doi: 10.1097/MCC.0000000000001029. PMID: 36880556.
Stefanou Ε, Karvelas N, Bennett S, Kole C. Cerebrovascular Manifestations of SARS-CoV-2: A Comprehensive Review. Curr Treat Options Neurol. 2023;25(4):71-92. doi: 10.1007/s11940-023-00747-6. Epub 2023 Mar 4. PMID: 36950279; PMCID: PMC9984763.
Moderate editing of English language required
Author Response
Comment: Before moving to review the manuscript, a graphical abstract is highly recommended, to under in one look.
Response: In this revision, we have provided a graphical abstract as requested.
Comment: Being a review manuscript, I would recommend the author's to improve the article by including similar articles, from the link (https://pubmed.ncbi.nlm.nih.gov/?term=Cerebrovascular+in+COVID+19&filter=pubt.review&filter=years.2023-2023&sort=pubdate&size=100) or some articles are selected or search has been refined, please.
Response: We thank the reviewer for their feedback. There are currently 95 references in our review which encompass original research, systematic review, and narrative review publications on the topic of “Cerebrovascular Disease in COVID”.
After reviewing the 36 manuscripts which return from the PubMed search provided by the reviewer, 17 are unrelated to cerebrovascular disease in COVID-19 (e.g., manuscripts relating to cholinergic deficiency state, melatonin and aging, neuro-ophthalmological complications of COVID-19, fatty liver disease, headache, multiple sclerosis, near hanging, etc). Of the remaining articles, several are too specific to provide meaningful content to our review (e.g., the role of hypovitaminosis D in thrombotic complications, the role of flavonoids, etc). The relevant manuscripts on the topic of our review have been cited, and we believe the review is as comprehensive as it can be given the space limitations. Other than the comment below, if there is a specific reference the reviewer would like for us to include, please advise.
Comment: The following articles need the attention of author's:
Menezes, R. G., Alabduladhem, T. O., Siddiqi, A. K., Maniya, M. T., Al Dahlawi, A. M., Almulhim, M. W. A., ... & Almas, T. (2023). Cerebrovascular disease in COVID-19: a systematic review and meta-analysis. Le Infezioni in Medicina, 31(2), 140
Siegler, J. E., Dasgupta, S., Abdalkader, M., Penckofer, M., Yaghi, S., & Nguyen, T. N. (2023). Cerebrovascular Disease in COVID-19.
Owens CD, Pinto CB, Detwiler S, Mukli P, Peterfi A, Szarvas Z, Hoffmeister JR, Galindo J, Noori J, Kirkpatrick AC, Dasari TW, James J, Tarantini S, Csiszar A, Ungvari Z, Prodan CI, Yabluchanskiy A. Cerebral small vessel disease pathology in COVID-19 patients: A systematic review. Ageing Res Rev. 2023 Jul;88:101962. doi: 10.1016/j.arr.2023.101962. Epub 2023 May 22. PMID: 37224885; PMCID: PMC10202464.
Menezes RG, Alabduladhem TO, Siddiqi AK, Maniya MT, Al Dahlawi AM, Almulhim MWA, Almulhim HW, Saeed YAA, Alotaibi MS, Alarifi SS, Alkathiry AM, Almas T. Cerebrovascular disease in COVID-19: a systematic review and meta-analysis. Infez Med. 2023 Jun 1;31(2):140-150. doi: 10.53854/liim-3102-2. PMID: 37283635; PMCID: PMC10241400.
Finsterer J. Neurological Adverse Reactions to SARS-CoV-2 Vaccines. Clin Psychopharmacol Neurosci. 2023 May 30;21(2):222-239. doi: 10.9758/cpn.2023.21.2.222. PMID: 37119215; PMCID: PMC10157009.
Solomon IH, Singh A, Folkerth RD, Mukerji SS. What Can We Still Learn from Brain Autopsies in COVID-19? Semin Neurol. 2023 Apr;43(2):195-204. doi: 10.1055/s-0043-1767716. Epub 2023 Apr 6. PMID: 37023787.
Sonneville R, Dangayach NS, Newcombe V. Neurological complications of critically ill COVID-19 patients. Curr Opin Crit Care. 2023 Apr 1;29(2):61-67. doi: 10.1097/MCC.0000000000001029. PMID: 36880556.
Stefanou Ε, Karvelas N, Bennett S, Kole C. Cerebrovascular Manifestations of SARS-CoV-2: A Comprehensive Review. Curr Treat Options Neurol. 2023;25(4):71-92. doi: 10.1007/s11940-023-00747-6. Epub 2023 Mar 4. PMID: 36950279; PMCID: PMC9984763.
Response: In the revision, we have included relevant content from several of these citations. Not all were included due to content from these manuscripts exceeding the scope of our present review, and others (like the citation of our review on a preprint server) which cannot be added to the manuscript. The Menezes article is mentioned twice by the reviewer here, and provides a meta-analysis of stroke event rates in COVID-19. We have added their findings in the section where the earlier Yamakawa meta-analysis was included.
Reviewer 2 Report
The topic is of great clinical interest and the review is an appreciable update on this topic. I think only some minor considerations are necessary:
Lines 243-246
“... Furthermore, there are presently no guidelines or societal recommendations for these scenarios.”
Actually, for example, the NIH provides constantly updated guidelines with specific recommendations on the topic in question ("Coronavirus Disease 2019 (COVID-19) Treatment Guidelines", https://www.covid19treatmentguidelines.nih.gov, updated on antithrombotic therapy on april 20, 2023).
Lines 250-252
“In patients with stroke and an indication for therapeutic anticoagulation (e.g., non-valvular atrial fibrillation), it is reasonable to treat with therapeutic anticoagulation”
It would be correct to eliminate the term "non-valvular" since also the “valvular” atrial fibrillation has an indication to for anticoagulant therapy.
Regarding Figure 2, I think it should be made clearer and more comprehensive, for example:
In critically and not critically ill patients with COVID 19 the part of the antithrombotic treatment algorithm in patient with no alternative indication to an OAC neither and the risk of anticoagulation does not outweight potential benefit is missing.
Any significant deviation of the antithrombotic treatment algorithm from that indicated by the NIH should be carefully considered, reported and justified also considering the potential implications of a forensic nature.
Still with regard to figure 2, although it may seem obvious, it is necessary to add that patients with COVID-19 and alternative indication to anticoagulant, when safe to anticoagulate, after a short term therapeutic anticoagulation with heparin, must have chronic anticoagulant therapy.
A final consideration concerns the definition of "critically and not critically ill patient with COVID-19" which should be reported to allow the reader a clinical application of the indications provided.
Just a minor editing is necessary
Author Response
Comment: The topic is of great clinical interest and the review is an appreciable update on this topic. I think only some minor considerations are necessary:
Lines 243-246
“... Furthermore, there are presently no guidelines or societal recommendations for these scenarios.”
Actually, for example, the NIH provides constantly updated guidelines with specific recommendations on the topic in question ("Coronavirus Disease 2019 (COVID-19) Treatment Guidelines", https://www.covid19treatmentguidelines.nih.gov, updated on antithrombotic therapy on april 20, 2023).
Response: We have included a reference to the NIH guidelines on COVID-19 treatment in the revision as follows, with changes to the manuscript bolded here:
“Furthermore, guidelines are limited for these scenarios. The 2021 American Heart Association guidelines on secondary stroke prevention (published 15 months after the World Health Organization declared COVID-19 a pandemic) do not mention COVID-19.57 Presumably, the risk of subsequent arterial or venous thrombosis following ischemic infarction in a patient with COVID-19 (particularly a non-critically patient with elevated d-dimer) is high.58,59 The National Institutes of Health COVID-19 Treatment Guidelines Panel most recently updated their recommendations in December 2022 for the treatment of COVID-19 and thromboembolic disease, but there is no specific mention of secondary stroke prevention in these guidelines.60 (Much of these recommendations pertain to the treatment of venous thromboembolism.)
Comment: Lines 250-252
“In patients with stroke and an indication for therapeutic anticoagulation (e.g., non-valvular atrial fibrillation), it is reasonable to treat with therapeutic anticoagulation”
It would be correct to eliminate the term "non-valvular" since also the “valvular” atrial fibrillation has an indication to for anticoagulant therapy.
Response: We have removed “non-valvular” as suggested.
Comment: Regarding Figure 2, I think it should be made clearer and more comprehensive, for example:
In critically and not critically ill patients with COVID 19 the part of the antithrombotic treatment algorithm in patient with no alternative indication to an OAC neither and the risk of anticoagulation does not outweight potential benefit is missing.
Any significant deviation of the antithrombotic treatment algorithm from that indicated by the NIH should be carefully considered, reported and justified also considering the potential implications of a forensic nature.
Still with regard to figure 2, although it may seem obvious, it is necessary to add that patients with COVID-19 and alternative indication to anticoagulant, when safe to anticoagulate, after a short term therapeutic anticoagulation with heparin, must have chronic anticoagulant therapy.
Response: We have updated the Figure as recommended and clarified it further. We have also referred the reader to the NIH recommendations and any societal guidelines that may be published in the coming months and years. Thank you.
Newly added:
“The reader is referred to the National Institutes of Health60 for the latest updates to antithrombotic recommendations in COVID-19, and to be cognizant of any future societal recommendations for the treatment of COVID-19-associated cerebrovascular disease. As with any off-label treatment, deviations from such recommendations or guidelines ought to be carefully considered and justified given the rapidly changing evidence for treatment.“
Comment: A final consideration concerns the definition of "critically and not critically ill patient with COVID-19" which should be reported to allow the reader a clinical application of the indications provided.
Response: We have defined “critically” ill in the revised caption of Figure 2 and cited it in the text of the manuscript:
“Definitions for “critically ill” and “non-critically ill” vary between studies. Generally speaking, “critically ill” patients with COVID-19 are those who require intensive care (e.g., requiring invasive mechanical ventilation, frequent nursing assessment of vital signs, use of vasopressors, or those with multi-organ failure) or are at high-risk of imminent deterioration (e.g., those with increasing oxygen requirements or with evidence of new or progressive organ dysfunction). Non-critically ill patients are those who are managed in a non-intensive medical or surgical unit, or those well enough to remain outpatient.”
Round 2
Reviewer 1 Report
Reply to the comments: Appreciated
Check spell of names etc